# *Schinus terebinthifolius* Leaf Lectin (SteLL) Reduces the Bacterial and Inflammatory Burden of Wounds Infected by *Staphylococcus aureus* Promoting Skin Repair

**DOI:** 10.3390/ph15111441

**Published:** 2022-11-21

**Authors:** Marcio Anderson Sousa Nunes, Lucas dos Santos Silva, Deivid Martins Santos, Brenda da Silva Cutrim, Silvamara Leite Vieira, Izadora Souza Soeiro Silva, Simeone Júlio dos Santos Castelo Branco, Mayara de Santana do Nascimento, André Alvares Marques Vale, Ana Paula Silva dos Santos-Azevedo, Adrielle Zagmignan, Joicy Cortez de Sá Sousa, Thiago Henrique Napoleão, Patrícia Maria Guedes Paiva, Valério Monteiro-Neto, Luís Cláudio Nascimento da Silva

**Affiliations:** 1Rede de Biodiversidade e Biotecnologia da Amazônia Legal, BIONORTE, São Luís 65055-310, Brazil; 2Laboratório de Patogenicidade Microbiana, Universidade Ceuma, São Luís 65075-120, Brazil; 3Instituto de Ciências Biomédicas, Universidade de São Paulo, São Paulo 05508-000, Brazil; 4Laboratório de Bioquímica de Proteínas, Centro de Biociências, Universidade Federal de Pernambuco, Recife 50740-570, Brazil; 5Centro de Ciências Biológicas e da Saúde, Universidade Federal do Maranhão, São Luís 65080-805, Brazil

**Keywords:** anti-infective properties, bacterial infection, Brazilian pepper tree, growth factors, plant lectins, wound contraction

## Abstract

*Staphylococcus aureus* is commonly found in wound infections where this pathogen impairs skin repair. The lectin isolated from leaves of *Schinus terebinthifolius* (named SteLL) has antimicrobial and antivirulence action against *S. aureus*. This study evaluated the effects of topical administration of SteLL on mice wounds infected by *S. aureus*. Seventy-two *C57/BL6* mice (6–8 weeks old) were allocated into four groups: (i) uninfected wounds; (ii) infected wounds, (iii) infected wounds treated with 32 µg/mL SteLL solution; (iv) infected wounds treated with 64 µg/mL SteLL solution. The excisional wounds (64 mm^2^) were induced on the dorsum and infected by *S. aureus* 432170 (4.0 × 10^6^ CFU/wound). The daily treatment started 1-day post-infection (dpi). The topical application of both SteLL concentrations significantly accelerated the healing of *S. aureus*-infected wounds until the 7th dpi, when compared to untreated infected lesions (reductions of 1.95–4.55-fold and 1.79–2.90-fold for SteLL at 32 µg/mL and 64 µg/mL, respectively). The SteLL-based treatment also amended the severity of wound infection and reduced the bacterial load (12-fold to 72-fold for 32 µg/mL, and 14-fold to 282-fold for 64 µg/mL). SteLL-treated wounds show higher collagen deposition and restoration of skin structure than other groups. The bacterial load and the levels of inflammatory markers (IL-6, MCP-1, TNF-α, and VEGF) were also reduced by both SteLL concentrations. These results corroborate the reported anti-infective properties of SteLL, making this lectin a lead candidate for developing alternative agents for the treatment of *S. aureus*-infected skin lesions.

## 1. Introduction

Several factors can cause a skin wound. Some are from medical origins (such as surgical procedures) and others are the result of accidents or illnesses. Regardless of the cause, these injuries affect millions of people around the world and challenge the health system [1,2,3]. Wound healing is a fundamental physiological process for preserving epithelial integrity and protecting the body. In normal conditions, tissue repair develops in four steps: hemostasis, inflammation, proliferation, and tissue remodeling [4,5,6].

Hemostasis begins immediately after the trauma, in which there is the activation of the coagulation cascade resulting from platelet growth factors and thrombin complex. Simultaneously, during the inflammation phase, the migration of neutrophils, macrophages, and other immune cells that are responsible for eliminating opportunistic pathogens occurs [7]. Then, neutrophils interact with components of the extracellular matrix (ECM), stimulating the formation of angiogenesis promoters, such as the basic fibroblast growth factor (bFGF) and vascular endothelial growth factor (VEGF) [8,9].

The proliferative phase begins with epithelial cells and fibroblasts that are recruited and assist in hair growth, collagen production, and formation of new tissue [5]. Consequently, granulation tissue is formed as the result of the formation of new capillaries and lymphatic vessels (angiogenesis). Finally, in the remodeling phase, the newly formed tissue is constantly remodeled until its characteristics approximate that of normal tissue [10].

One of the main factors that compromise healing is the occurrence of infections both by microorganisms of the skin microbiota and by the acquisition of others present in the environment [11,12,13]. *Staphylococcus aureus* is among the bacterial species most found in superficial infections, whose prevalence is related to its presence in skin microbiota, the vast arsenal of virulence determinants, and high antibiotic resistance [14,15]. The virulence factors produced by *S. aureus* include coagulase (Coa), factor-binding protein von Willibrand (vWbp), alpha-hemolysin and phenol-soluble modulins (PSMs), and biofilm synthesis [16,17].

Due to the difficulties faced by health systems in the treatment of skin infections, the search for new bioactive compounds with antimicrobial and/or healing properties is increasing [18]. Lectins, a group of proteins widely distributed in nature that reversibly binds to carbohydrates, present several biotechnological applications including anti-infective, immunomodulatory, and healing actions [19,20,21].

An example is SteLL, a lectin isolated from the leaves of *Schinus terebinthifolius* (Anacardiaceae), a plant widely used in Brazilian folk medicine [22,23]. SteLL is characterized as an *N*-acetylglucosamine binding protein with a molecular weight of 12.4 kDa and it presents a promising antibacterial compound inhibiting the growth of bacteria of clinical interest, including *S. aureus* [24,25]. Furthermore, its immunomodulatory action was described in different works, and its anti-infection activity in experimental models using mice macrophages and *Galleria mellonella* larvae [25]. SteLL also has antitumor, analgesic, and anxiolytic activities in mice bearing sarcoma 180 [26,27,28,29].

Given the anti-infective and immunomodulatory properties of SteLL, this study aimed to evaluate the effects of its topical administration in an infection model of wound infection caused by *S. aureus*. The null hypotheses are that SteLL-based treatment (1) would not present antimicrobial effects in the in vivo model of *S. aureus*-infected wounds; (2) would not reduce the severity of wound inflammation; (3) would not promote wound healing and skin repair; and (4) and would not present in vivo immunomodulatory effects.

## 2. Results

### 2.1. Topical Treatment with SteLL Reduced the Severity of S. aureus-Infected Skin Wounds

During the 17 days of the experiment, the macroscopy aspects (wound area, amount and type of exudate, edema intensity, color of surrounding skin tissue, type of debridement tissue) of each wound were evaluated daily for determination of an index of severity. In the uninfected control, due to the absence of an infectious agent, the inflammatory parameters were limited to serous-type exudate which decreased from moderate to mild as the lesion closed, and granulation tissue was observed after five days post-infection (dpi). The higher degrees of severity were observed during the first five dpi in the infected group (Figure 1 and Figure 2). These wounds show a high amount of purulent exudate and edema, the main indicators of intense inflammatory response. Furthermore, the formation of granulation tissue was observed only on the tenth day (Figure 1).

In the groups of animals treated with both SteLL concentrations, moderate edema and moderate serous exudate were observed. Compared to the infected group, the groups treated with SteLL had a lower degree of severity, which indicate that the presence of the lectin decreased the need for a more intense inflammatory response, besides having presented granulation tissue from the 7th dpi.

The index of severity is represented in Figure 2. It is possible to observe that the infected group had the highest scores of infection severity (Figure 2a). On the other hand, treatment with SteLL (in both tested concentrations) reduced the severity of infection from the seventh and eighth day of treatment at the concentration of 64 µg/mL and 32 µg/mL, respectively, presenting only mild serous exudate and granulation tissue.

These data are more evident using the area under the curve (AUC) analysis (Figure 2b), where the infected group had the highest mean AUC (72.28 ± 9.21 mm^2^) and the uninfected had the lowest mean AUC (34.0 ± 9.21 mm^2^). Indeed, the AUC values for SteLL-treated groups were lower than infected animals (AUC of 47.25 ± 7.54 mm^2^ and 48.39 ± 7.62 mm^2^ for 32 µg/mL and 64 µg/mL, respectively; *p* < 0.0001).

### 2.2. Topical Treatment with SteLL Accelerated the Contraction of S. aureus-Infected Skin Wounds

Regarding the wound area (Figure 3), the values were expressed in percentage (%), where the area at 1st dpi represents 100%. Concerning the uninfected groups, the infected wounds without treatment show higher areas until the 7th dpi. Similarly, the animals treated with SteLL show lower wound areas until the 7th dpi in relation to the infected group (*p* < 0.05). Animals treated with SteLL at 32 µg/mL show reductions ranging from 1.95–4.55-fold concerning untreated infected wounds, while for SteLL at 64 µg/mL the reduction ranged from 1.79–2.90-fold. The groups treated with SteLL at concentrations of 32 µg/mL and 64 µg/mL show no significant statistical differences (*p* > 0.05) among themselves and in relation to uninfected animals.

To better characterize the efficiency of topical treatment with SteLL, histological analyses were performed using Hematoxylin and Eosin (HE) and Masson’s trichrome staining (Figure 4, Figure 5, Figure 6, Figure 7 and Figure 8; Table 1). At the 3rd dpi, all infected wounds show high amounts of cell debris and intense inflammatory infiltrate (with a predominance of polymorphonuclear cells), although slightly lower scores were recorded for those treated with SteLL at 64 µg/mL (*p* < 0.05). After 10 days of treatment, cell debris and moderate inflammatory infiltrate were present in untreated infected wounds and those treated with SteLL at 64 µg/mL. The group treated with SteLL at 32 µg/mL showed improved re-epithelization and mild inflammatory infiltrate (*p* < 0.05).

Following, at the 17th dpi, inflammatory infiltrate and cell debris were not detected in the uninfected wounds and SteLL-treated groups. The epidermis presented patterns of normality similar to the control group; that is, it is characterized by the strata (basal, spinous, granular, and corneal). The dermis of these animals shows high cellularity (fibroblasts), uniform distribution of collagen fiber bundles, and a reduction in the inflammatory infiltrate, especially after 17 days of treatment. The dermis shows a more uniform and denser distribution of collagen fiber bundles in SteLL-treated groups. On the other hand, the presence of cell debris and inflammatory infiltrate was still verified in the untreated infected group.

### 2.3. Topical Treatment with SteLL Reduced the Bacterial Load of Skin Lesions Infected by S. aureus

The in vivo antimicrobial effects of the topical administration of SteLL were evaluated at 3 dpi, 10 dpi, and 17 dpi (Figure 9). When compared to uninfected animals, the wounds infected by *S. aureus* presented CFU values increased by approximately 1683-fold, 117-fold, and 68-fold after 3 dpi (Figure 9a), 10 dpi (Figure 9b), and 17 dpi (Figure 9c) (*p* < 0.0001). On the other hand, the topical treatment with both SteLL concentrations resulted in statistical differences in bacterial load in relation to untreated infected wounds (*p* < 0.0001). Specifically, the administration of the lectin at 32 µg/mL resulted in reductions in bacterial load of the order of 12-fold (3 days of treatment), 72-fold (10 days of treatment), and 12-fold (17 days of treatment). Similarly, mice treated with SteLL at 64 µg/mL show values of *S. aureus* load decreased by 14-fold, 282-fold, and 14-fold in the tested periods.

### 2.4. Topical Treatment with SteLL Modulated the Levels of Inflammatory Markers at Skin Lesions Infected by S. aureus

The levels of inflammatory markers (IFN-γ, IL-6, IL-10, IL-12p70, MCP-1, TNF-α, and VEGF) were measured in the lesion tissue on the 3rd dpi, and its production was reported in picograms per mg of protein (pg/mg) (Figure 10). The infection induced by *S. aureus* significantly increased the production c IL-6, MCP-1, TNF-α, and VEGF in relation to the lesions without infection (*p* < 0.001); for other markers, no conclusive results were obtained. On the other hand, the topical application of SteLL in *S. aureus*-infected wounds reduced the amounts of these markers to values similar to those detected in animals without infection (*p* > 0.05). The average reductions for both SteLL concentrations were 97.41% for IL-6 (*p* < 0.0001), 97.87% for MCP-1 (*p* < 0.0001), 87.09% for TNF-α (*p* < 0.0001) and 41.40% for VEGF (*p* < 0.01).

## 3. Discussion

*S. aureus* is commonly found in wound infections where this pathogen increases the time for healing and skin repair [17,30]. In this context, there is an urgent need for compounds able to treat wound infection by direct inhibition of bacterial growth or by improving the host response (immune system and healing process) [31,32,33,34]. Some lectins were highlighted as promising healing agents for skin wounds, especially those with an affinity to mannose [32,35] and galactose [36,37]. Despite the immunomodulatory, healing, and anti-infective potential of these proteins, their application on models of skin infection has not been properly addressed [32]. Herein, the effects of the topical administration of SteLL, an *N*-acetylglucosamine-binding protein, were evaluated in a murine model of skin wounds infected by *S. aureus*.

In addition to these previous results, the topical treatment with both SteLL concentrations (32 µg/mL and 64 µg/mL) significantly decreased the bacterial load in the tissue, reducing the inflammation severity and promoting faster wound healing. Indeed, SteLL is reported as a protein with the ability to inhibit the growth of Gram-negative and Gram-positive bacteria of clinical interest, having a stronger inhibition of *S. aureus* [14], which is associated with the fact that this lectin specifically binds to *N*-acetylglucosamine [15,16], a carbohydrate present in the cell wall of Gram-positive bacteria [17]. The in vivo antimicrobial action of SteLL was previously demonstrated using *G. mellonella*. The larvae infected by *S. aureus* and treated with SteLL had a longer survival compared to untreated larvae, with a decrease in bacterial load in hemolymph [25].

During the experiment, the wound contraction and the clinical characteristics (as markers of inflammation severity) were evaluated. The SteLL-treated animals show faster wound contraction and a reduced severity score when compared to untreated animals. These results are confirmed by the lower detection of cytokines and VEGF in mice treated with SteLL in comparation to other groups. The healing action of SteLL seems to be related to its antimicrobial and immunomodulatory effects since no significant healing effects were observed in uninfected wounds treated with the tested SteLL concentrations (unpublished data).

The immunomodulatory effects of SteLL were reported in mice macrophages (infected or not by *S. aureus*) and mice splenocytes [25,38]. In uninfected macrophages, the lectin stimulated the production of mitochondrial superoxide, nitric oxide, and cytokines (IL-6, IL-10, IL-17A, and TNF-α). SteLL-treated macrophages show improved bactericidal action toward *S. aureus* and reduced expression of IL-17A and IFN-γ in infected macrophages [25]. The immunomodulatory effects of SteLL were also evaluated in mice splenocytes. These cells when treated with SteLL show an increased release of pro-inflammatory cytokines (IL-2, IL-17A, IFN-γ, and TNF-α) and anti-inflammatory cytokines (IL-4) [38].

The creation of a new vasculature is an essential phase of the healing pathway to promote the delivery of nutrients, oxygen, immune cells, inflammatory mediators, and growth factors [39]. In general, angiogenesis is correlated with the inflammatory response, due to the production of proangiogenic mediators by inflammatory cells [13,40]. In this sense, the excessive inflammation could lead to increased creation of a dense but poorly perfused capillary bed which could result in lower healing outcomes [40,41].

In this context, as seen in our results, the severe inflammation response trigged by *S. aureus* results in increased levels of VEGF, one of the major growth factors associated with angiogenesis [42,43]. Importantly, the SteLL treatment significantly reduced the VEGF levels of *S. aureus*-infected wounds. The values were similar to those detected in uninfected wounds. The antiangiogenic properties of SteLL were previously observed in sarcoma 180-bearing mice and associated with its antitumor action [26].

As described above, the healing actions of some lectins were previously demonstrated such as those isolated from *Cratylia mollis* (Cramoll) [32], *Parkia Pendula* [35], *Artocarpus incisa* (frutalin) [36], and *Bauhinia variegata* (BVL) [37]. However, only Cramoll, a mannose-binding lectin was reported as an immunomodulatory agent in *S. aureus*-infected skin wounds [32]. In addition, to the best of our knowledge, it is the first report about the in vivo antimicrobial activity of an *N*-acetylglucosamine-binding lectin. However, a limitation of this present study is that the role of the sugar-binding domain in the action of SteLL was not determined, as well as the molecular mechanisms involved.

## 4. Materials and Methods

### 4.1. SteLL Purification

Leaves of *S. teribinthifolius* were collected in Recife (Brazil) at the campus of the Universidade Federal de Pernambuco. The lectin purification was performed using the methodology previously described [24]. In short, the powder obtained from the dried leaves (20 g) was diluted in saline solution (0.15 M NaCl) and subjected to agitation (4 °C). After 16 h, the filtered extract was centrifuged (3000 rpm/15 min) and passed through activated carbon (10%), and then subjected to a chitin column (Sigma-Aldrich, St. Louis, MO, USA). After elution, SteLL was obtained after dialysis (10 kDa cut-off membrane; Sigma-Aldrich, St. Louis, MO, USA) with distilled water (6 h, 4 °C). Protein concentration was determined using a standard curve of bovine serum albumin (BSA, Sigma-Aldrich, St. Louis, MO, USA) [44]. The hemagglutinating activity (HA) assay was performed to ensure the lectin functionality [45]. SteLL homogeneity was confirmed by polyacrylamide gel electrophoresis (SDS-PAGE) under denaturing conditions.

### 4.2. Animals and Ethical Conditions

The study was carried out in the vivarium of CEUMA University in São Luís (MA), after approval by the Ethics Committee for the Use of Animals of the institution (CEUA-UNICEUMA), under Protocol No. 00013/18. Seventy-two healthy *C57*/*BL6* mice (6 to 8 weeks) were used in this study and presented body weights between 19 to 30 g. The animals were distributed in four groups (as described below) and housed in cages of polypropylene placed in a ventilated rack with independent insufflation and exhaust systems to reduce the risk of contamination. The experimental procedures were conducted in an airy room with an average temperature of 21 °C and a 12 h light-dark cycle. During all periods, water and food were served ad libitum.

### 4.3. Induction of Experimental Skin Injuries and Treatment

The animals were anesthetized by intramuscular administration of xylazine hydrochloride (1 mg/kg) and ketamine chloride (50 mg/kg). After, in a sterile environment, the dorsal thoracic region was trichotomized and cleaned with 70% ethyl alcohol and sterile saline solution (150 mM NaCl). The wound area was demarcated (64 mm^2^) and the skin was removed with blunt-tipped scissors and dissection forceps. The wound infection was performed by *S. aureus* 432170 (4.0 × 10^6^ CFU/wound), an isolate obtained from the foot ulcer of a patient with type 2 diabetes mellitus [46].

The animals were separated by group: (1) mice with uninfected wounds and daily treated with 50 µL of PBS (uninfected control group); (2) mice with wounds infected by *S. aureus* and daily treated with 50 µL of PBS (infected control group); (3) mice with wounds infected by *S. aureus* and daily treated with 50 µL of SteLL at 32 µg/mL; (4) mice with wounds infected by *S. aureus* and daily treated with 50 µL of SteLL at 64 µg/mL. The treatment started 24 h after the infection. Six animals of each group were euthanized by anesthetic overdose at 3rd dpi, 10th dpi, and 17th dpi, and the skin fragments (covering the lesion area and intact skin) were collected for histological analysis and quantification of bacterial load and inflammatory mediators (Figure 11).

### 4.4. Analysis of Macroscopic Aspects

The wounds were photographed daily, and their area was calculated using the Image J program (National Institutes of Health) [47]. The macroscopic analysis of the wounds was performed within a laminar flow to calculate a severity score based on the following clinical parameters: wound area (0–7), amount of exudate (0–3), type of exudate (0–4), edema intensity (0–3), the color of surrounding skin tissue (0–4), type of debridement tissue (0–3). The score of each animal was recorded daily until the end of the experiment [48].

### 4.5. Histological Evaluation

The skin fragments were fixed in 10% buffered formaldehyde (pH 7.2) and tissue sections were obtained (3–5 µm) that were stained with Hematoxylin and Eosin and Masson’s trichrome staining. The material was analyzed by light microscopy (at least 10 fields). The criteria evaluated included cellular debris, inflammatory infiltrate, reepithelization, vascularization, fibroblast proliferation, and distribution pattern of collagen fibers. The following scores were established for the intensity of histological findings: absent (when the parameter was not observed in any field), discrete (when observed in 1 to 3 fields), moderate (when observed in 4 to 6 fields), and intense (when observed in over 7 fields).

### 4.6. Bacterial Quantification

The quantification of viable bacteria from the skin lesions was performed on the 4th dpi, 11th dpi, and 17th dpi. The collected tissues were macerated with sterile PBS using a vortex (five cycles of 30 s) followed by centrifugation (5 min at 2500 RPM). The resulting solution was 10-fold diluted and plated on Mannitol Salt Agar. Each plate was incubated for 24 h at 37 °C and the bacterial load was expressed as CFU/g of tissue.

### 4.7. Quantification of Inflammatory Markers

Cytokines (IFN-γ, IL-6, IL-10, IL-12p70, MCP-1, TNF-α) in wound tissues were measured using Mouse Inflammation Kit BD^TM^ Cytometric Bead Array (CBA; BD Biosciences, São Paulo, Brazil). The analysis was performed in a BD Accuri C6 flow cytometer, according to the manufacturer’s instructions. The results were obtained using CBA FCAP Array software (BD Biosciences, São Paulo, Brazil) and expressed as pg/g of tissue.

The levels of VEGF were determined by the Mouse VEGF ELISA Kit (Sigma-Aldrich; São Paulo, Brazil), following the manufacturer’s instructions. The protein concentration in each sample was quantified by Bradford reagent (Sigma-Aldrich; São Paulo, Brazil) using a standard curve of BSA. The absorbance values were obtained using a spectrophotometer (Plate reader MB-580; Heales, Shenzhen, China). The results were expressed as VEGF per mg of total protein.

### 4.8. Statistical Analysis

Results were expressed as mean ± standard error. The inhibition percentages were calculated as the average of the values obtained for each experiment. The graphs and statistical evaluation of the results were performed in Graphpad Prism 5.0. The data were analyzed by One-Way Analysis of Variance (ANOVA), followed by the Bonferroni test (GraphPad Software Inc., San Diego, CA, USA). A significance level lower than 0.05 was adopted.

## 5. Conclusions

The topical administration of SteLL reduced the bacterial load at wound sites, decreasing the inflammatory severity due to the lower release of inflammatory markers (IL-6, MCP-1, TNF-α, and VEGF). These effects led to faster wound contraction and improved skin repair. The data provide more insight into the in vivo antimicrobial and immunomodulatory effects of SteLL. To the best of our knowledge, it is the first evaluation of the in vivo antimicrobial activity of an *N*-acetylglucosamine-binding lectin.

Taken together, the findings of this study indicate that SteLL has great potential for use in the development of agents for wound healing. Future studies are needed to elucidate the molecular mechanisms involved in SteLL healing effects. In addition, other projects are in progress to design new dressing incorporated with SteLL to be evaluated in vivo models.

## Figures and Tables

**Figure 1 pharmaceuticals-15-01441-f001:**
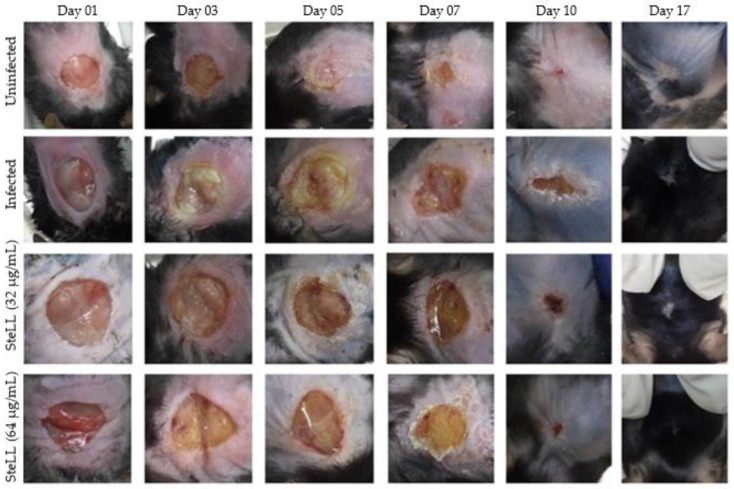
Macroscopic analysis of the effect of topical SteLL treatment (32 µg/mL and 64 µg/mL) on skin lesions infected by *Staphylococcus aureus*.

**Figure 2 pharmaceuticals-15-01441-f002:**
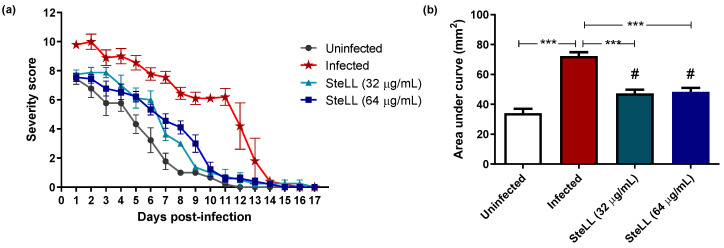
Effects of topical treatment with SteLL on the severity of wounds infected by *Staphylococcus aureus*. (**a**) Analysis of the clinical parameters. (**b**) Area under the curve (AUC) of the clinical parameters of the animals studied. *** Statistical differences among the indicated experimental groups (*p* < 0.0001); # Statistical differences in relation to uninfected group (*p* < 0.0001).

**Figure 3 pharmaceuticals-15-01441-f003:**
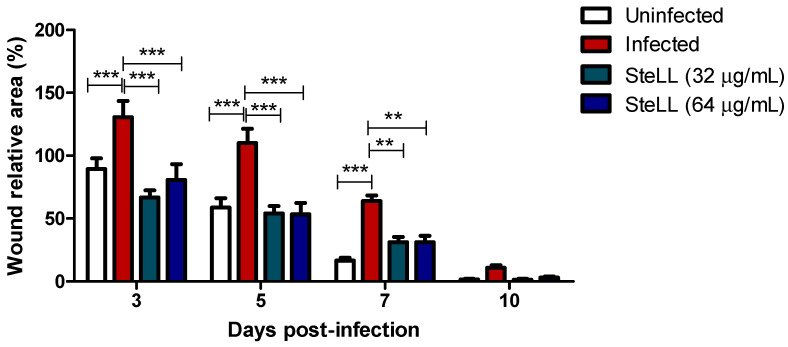
Effect of topical treatment with SteLL in the contraction of *Staphylococcus aureus*-infected wounds. Area expressed in percentage (%), where the lesion area on day one represents 100%. *** Statistical differences among the indicated experimental groups (*p* < 0.001); ** Statistical differences among the indicated experimental groups (*p* < 0.01).

**Figure 4 pharmaceuticals-15-01441-f004:**
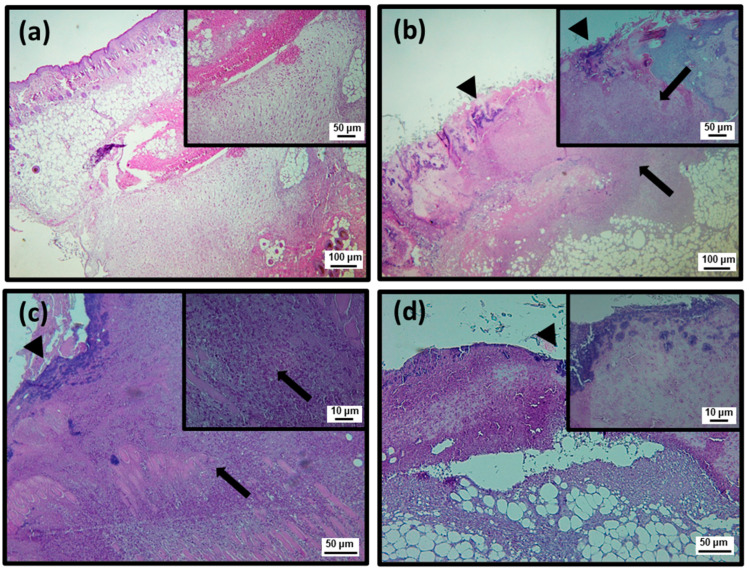
Histopathological analysis of healing process induced by 3 days of topical treatment with SteLL in *Staphylococcus aureus*-infected wounds using Hematoxylin and Eosin (HE) staining. Polymorphonuclear inflammatory infiltrate (arrow); Cell debris (arrowhead); (**a**) Uninfected; (**b**) Infected; (**c**) SteLL (32 μg/mL); (**d**) SteLL (64 μg/mL).

**Figure 5 pharmaceuticals-15-01441-f005:**
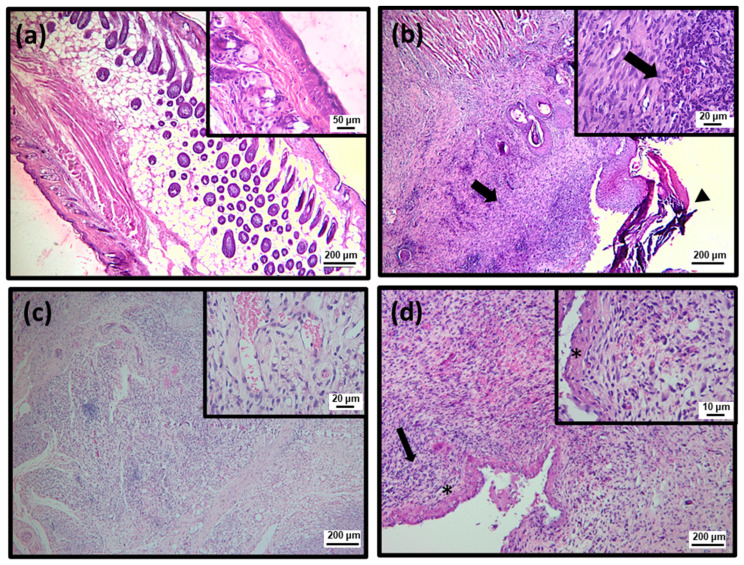
Histopathological analysis of healing process induced by 10 days of topical treatment with SteLL in *Staphylococcus aureus*-infected wounds using Hematoxylin and Eosin (HE) staining. Polymorphonuclear inflammatory infiltrate (arrow); Cell debris (arrowhead); Re-epithelialization (asterisk). (**a**) Uninfected; (**b**) Infected; (**c**) SteLL (32 μg/mL); (**d**) SteLL (64 μg/mL).

**Figure 6 pharmaceuticals-15-01441-f006:**
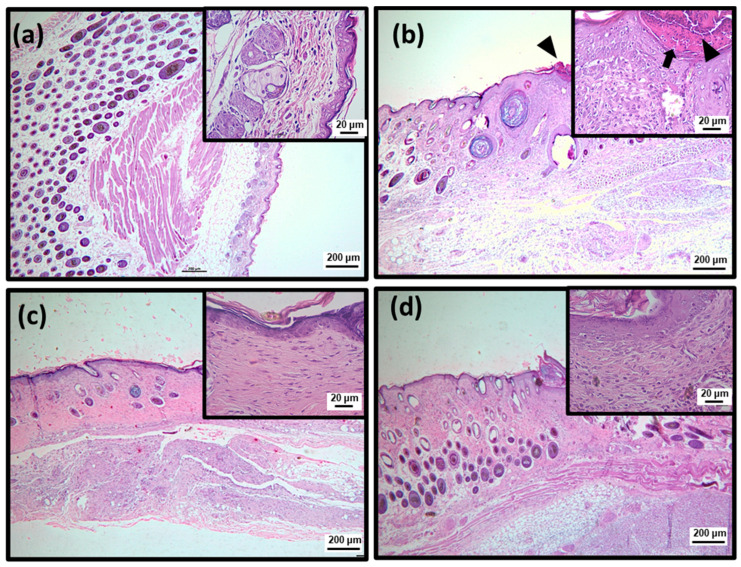
Histopathological analysis of healing process induced by 17 days of topical treatment with SteLL in *Staphylococcus aureus*-infected wounds using Hematoxylin and Eosin (HE) staining. Polymorphonuclear inflammatory infiltrate (arrow); Cell debris (arrowhead); (**a**) Uninfected; (**b**) Infected; (**c**) SteLL (32 μg/mL); (**d**) SteLL (64 μg/mL).

**Figure 7 pharmaceuticals-15-01441-f007:**
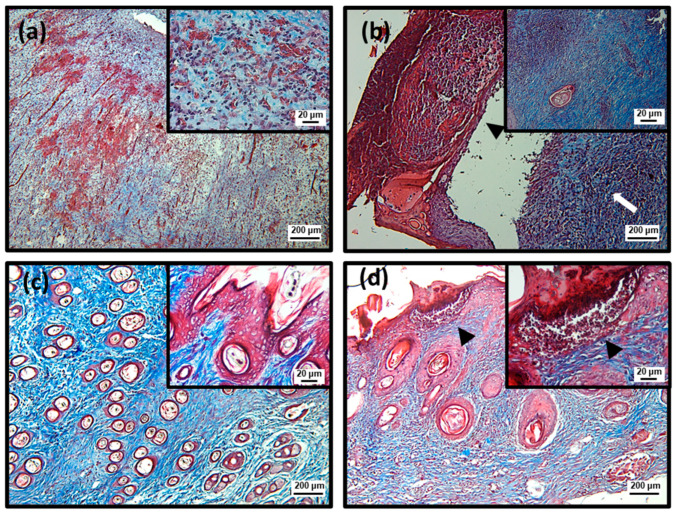
Histopathological analysis of the healing process induced by 10 days of topical treatment with SteLL in *Staphylococcus aureus*-infected wounds using Masson’s trichrome staining. Polymorphonuclear inflammatory infiltrate (arrow); Cell debris (arrowhead). The pictures are at magnifications of 40× (200 µm) and 400× (20 µm). (**a**) Uninfected; (**b**) Infected; (**c**) SteLL (32 μg/mL); (**d**) SteLL (64 μg/mL).

**Figure 8 pharmaceuticals-15-01441-f008:**
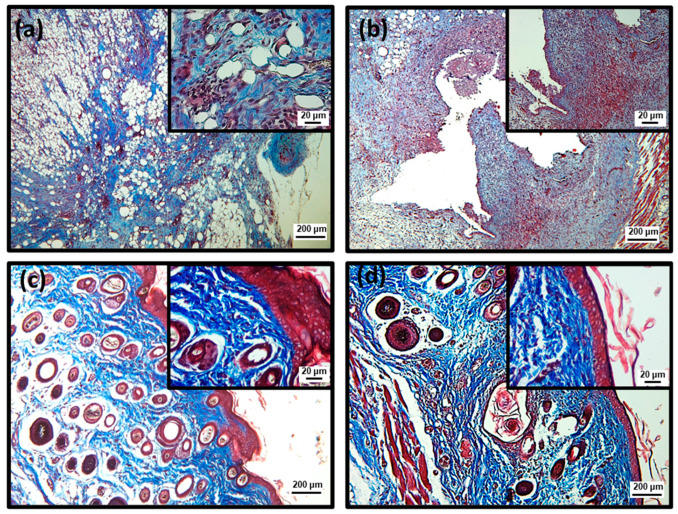
Histopathological analysis of the healing process induced by 17 days of topical treatment with SteLL in *Staphylococcus aureus*-infected wounds using Masson’s trichrome staining. (**a**) Uninfected; (**b**) Infected; (**c**) SteLL (32 μg/mL); (**d**) SteLL (64 μg/mL).

**Figure 9 pharmaceuticals-15-01441-f009:**
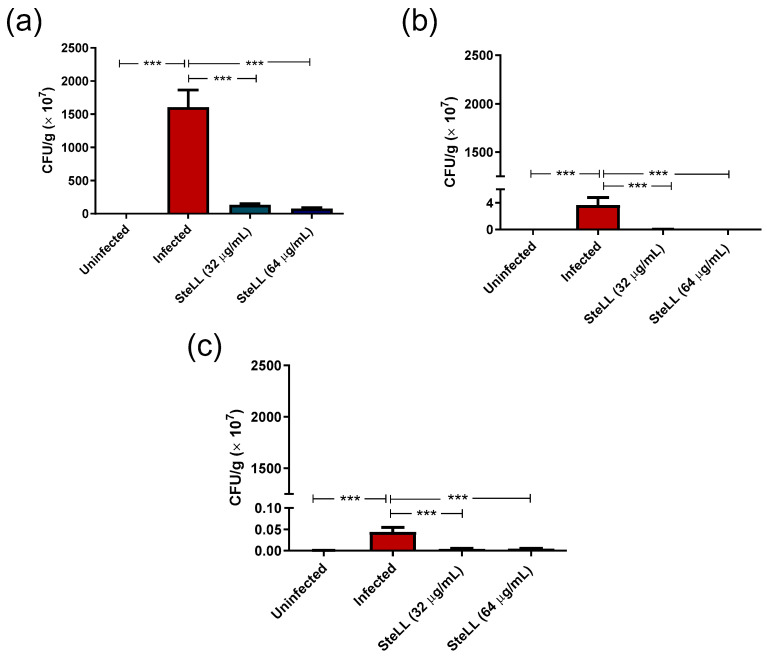
Effect of topical treatment with SteLL (32 µg/mL and 64 µg/mL) on bacterial load in wounds contaminated by *Staphylococcus aureus*. (**a**) Bacterial load after 3 days of infection; (**b**) Bacterial load after 10 days of infection; (**c**) Bacterial load after 17 days of infection. *** Statistical differences (*p* < 0.0001).

**Figure 10 pharmaceuticals-15-01441-f010:**
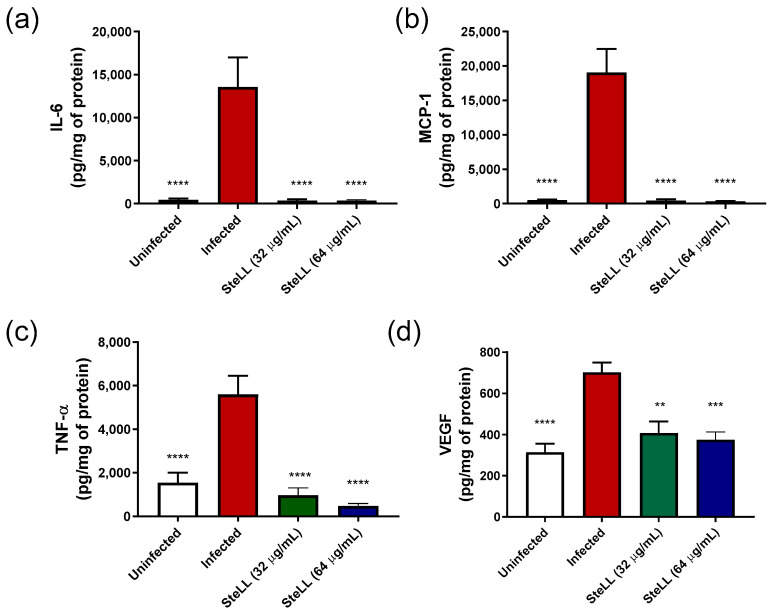
Effect of topical SteLL treatment on inflammatory markers levels in wound tissue infected by *Staphylococcus aureus.* (**a**) IL-6; (**b**) MCP-1; (**c**) TNF-α; (**d**) VEGF. ** *p* < 0.01; *** *p* < 0.001; **** *p* < 0.0001.

**Figure 11 pharmaceuticals-15-01441-f011:**
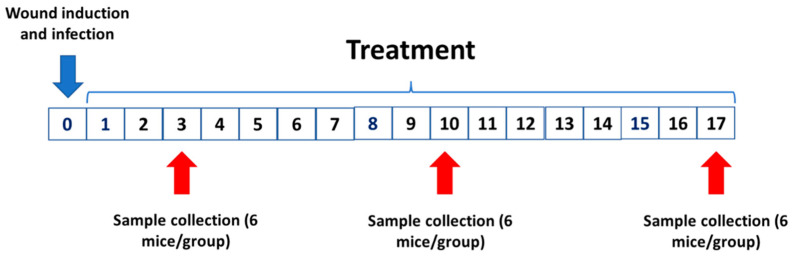
Experimental timeline followed in this study.

**Table 1 pharmaceuticals-15-01441-t001:** Histopathological evaluation of healing processes induced by topical treatment with SteLL in *Staphylococcus aureus*-infected wounds.

	Parameter	Uninfected	Infected	SteLL	
				32 µg/mL	64 µg/mL
3 days	Cell debris	2 ± 0 ^a^	3 ± 0 ^b^	3 ± 0 ^b^	2.25 ± 0.5 ^a^
	Re-epithelization	0 ± 0 ^a^	0 ± 0 ^a^	0 ± 0 ^a^	0 ± 0 ^a^
	Inflammatory infiltrate	1.25 ± 0.5 ^a^	3 ± 0 ^b^	3 ± 0 ^b^	2.5 ± 0.58 ^c^
	Fibroblast proliferation	-	-	-	-
10 days	Cell debris	0 ± 0 ^a^	1 ± 0 ^b^	0 ± 0 ^a^	1 ± 0.58 ^b^
	Re-epithelization	3 ± 0 ^a^	2.5 ± 0.5 ^b^	3 ± 0 ^a^	2.0 ± 0.5 ^c^
	Inflammatory infiltrate	0 ± 0 ^a^	2 ± 0 ^b^	1 ± 0 ^c^	2 ± 0 ^b^
	Fibroblast proliferation	1.25 ± 0.5 ^a^	3 ± 0 ^b^	3 ± 0 ^b^	2.5 ± 0.58 ^c^
17 days	Cell debris	0 ± 0 ^a^	0.5 ± 0 ^b^	0 ± 0 ^a^	0 ± 0 ^a^
	Re-epithelization	3 ± 0 ^a^	3 ± 0 ^a^	3 ± 0 ^a^	3 ± 0 ^a^
	Inflammatory infiltrate	0 ± 0 ^a^	1 ± 0 ^b^	0 ± 0 ^a^	0 ± 0 ^a^
	Fibroblast proliferation	2.25 ± 0.5 ^a^	2 ± 0 ^a^	3 ± 0 ^b^	3 ± 0 ^b^

In each line, different superscript letters (^a,b,c^) indicate significant difference (*p* < 0.05).

## Data Availability

Data is contained within the article.

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
