# Peer review of "Schinus terebinthifolius Leaf Lectin (SteLL) Reduces the Bacterial and Inflammatory Burden of Wounds Infected by Staphylococcus aureus Promoting Skin Repair"

_pharmaceuticals, 2022, doi:10.3390/ph15111441_

Round 1

Reviewer 1 Report

1. Title - need to revise to give better understanding

2. Abstract - 'Line 34 - need to mention the day of DPI

3. Abstract - 'Line 38 - need to correct the' restauration word'

4. Introduction- Line 58 - need to use proper abbreviation, which is ECM.

5. Introduction - need to highlight the significant od the study at last paragraph

6. Conclusion - some sentences need to be revised due to similarity with abstract.

Author Response

Dear reviewer,

Thank you so much for all suggestions which have improved the quality of our manuscript. Please find below a point-by-point response for your comments. The changes are highlighted in yellow in the updated version of our manuscript.

  1. Title - need to revise to give better understanding

Our response: We have changed the title for ‘Schinus terebinthifolius leaf lectin (SteLL) reduces the bacterial and inflammatory burden of wounds infected by Staphylococ-cus aureus promoting skin repair.’

  1. Abstract - 'Line 34 - need to mention the day of DPI

Our response: We have included.

  1. Abstract - 'Line 38 - need to correct the' restauration word'

Our response: We have corrected.

  1. Introduction- Line 58 - need to use proper abbreviation, which is ECM.

Our response: We have corrected.

  1. Introduction - need to highlight the significant od the study at last paragraph

Our response: We have included.

  1. Conclusion - some sentences need to be revised due to similarity with abstract.

Our response: We have corrected.

Reviewer 2 Report

This study evaluated the effects of the topical administration of SteLL 88 in an infection model of wound infection caused by S. aureus. This knowledge produced is relevant to the field of the Journal. This research is under the scope of this Journal.

There are some aspects which are possibly improved in the various sections of the manuscript:

  • Correct typos in all manuscripts.

-The use of personal pronouns should be avoided. Example “We have, we irrigated…etc”.

(Keywords)  

  • Please order the keywords / Mesh Terms alphabetically for a standardized presentation of the keywords.

(Abstract)

-  In the results, is important to show more information, and add some of the values.

(Introduction)

  • Identified the aim and null hypothesis at the end of the introduction.

(Materials and Methods)

  • When mentioning materials or devices: manufacturer, for some the manufacturer and city, for some you mention the manufacturer and city/ country. On Texas and California add also USA.
  • How was the sample calculated? Did the authors perform a power analysis to evaluate if this sample size was appropriate? What are the control groups?
  • How many operators performed the experiments?
  • Add the experimental time in the flowchart.

(Discussion)

- Compare your results with other emerging Antimicrobial peptides (AMPs), such as LL37 peptides, which may be immobilized nanoparticles to render them with antimicrobial and angiogenic properties. Please, read the Akhilesh Rai, 2021, 2022   https://doi.org/10.1039/D1BM01034D;https://doi.org/10.1039/D1TB02617H

-In  clarified some limitations of this study, and, add some future perspectives.

(References)

  • Check the reference’s MDPI format.

Author Response

Dear reviewer,

Thank you so much for all suggestions which have improved the quality of our manuscript. Please find below a point-by-point response for your comments. The changes are highlighted in yellow in the updated version of our manuscript.

This study evaluated the effects of the topical administration of SteLL 88 in an infection model of wound infection caused by S. aureus. This knowledge produced is relevant to the field of the Journal. This research is under the scope of this Journal.

There are some aspects which are possibly improved in the various sections of the manuscript:

  • Correct typos in all manuscripts. -The use of personal pronouns should be avoided. Example “We have, we irrigated…etc”.

Our response: We have corrected.

(Keywords)  

  • Please order the keywords / Mesh Terms alphabetically for a standardized presentation of the keywords.

Our response: We have corrected.

(Abstract)

  -  In the results, is important to show more information, and add some of the values.

Our response: We have included.

(Introduction)

  • Identified the aim and null hypothesis at the end of the introduction.

Our response: We have included.

(Materials and Methods)

  • When mentioning materials or devices: manufacturer, for some the manufacturer and city, for some you mention the manufacturer and city/ country. On Texas and California add also USA.

Our response: We have included.

  • How was the sample calculated? Did the authors perform a power analysis to evaluate if this sample size was appropriate? What are the control groups?

Our response: The sample size calculation was performed using the G Power software (available online: https://stats.idre.ucla.edu/other/gpower/), based on the standard error of similar experiments. We have included the designation of the control groups (uninfected and infected).

  • How many operators performed the experiments?

Our response: The in vivo assays were conducted by three operators.

  • Add the experimental time in the flowchart.

Our response: We have included.

(Discussion)

 - Compare your results with other emerging Antimicrobial peptides (AMPs), such as LL37 peptides, which may be immobilized nanoparticles to render them with antimicrobial and angiogenic properties. Please, read the Akhilesh Rai, 2021, 2022   https://doi.org/10.1039/D1BM01034D;https://doi.org/10.1039/D1TB02617H

Our response: Dear reviewer, we would like to thank you for this suggestion. However, as SteLL is a protein with native molecular mass of 12.4 kDa, the action mechanism should be different from small peptides such as LL37 (17-29 aa).

 -In  clarified some limitations of this study, and, add some future perspectives.

Our response: We have included

(References)

  • Check the reference’s MDPI format.

Our response: We have checked.

Round 2

Reviewer 2 Report

The authors improve the article with the reviewer comments